# Development of *N*-Acetylated Dipalmitoyl-*S*-Glyceryl Cysteine Analogs as Efficient TLR2/TLR6 Agonists

**DOI:** 10.3390/molecules24193512

**Published:** 2019-09-27

**Authors:** Yang Zhou, Abid H. Banday, Victor J. Hruby, Minying Cai

**Affiliations:** Department of Chemistry and Biochemistry, The University of Arizona, Tucson, AZ 85721, USA; yangzhou@email.arizona.edu (Y.Z.); abidrrl@gmail.com (A.H.B.); hruby@email.arizona.edu (V.J.H.)

**Keywords:** cancer vaccine, synthetic vaccine, adjuvant, Toll-like receptor, Pam_2_Cys, *N*-acetylated Pam_2_Cys

## Abstract

Cancer vaccine is a promising immunotherapeutic approach to train the immune system with vaccines to recognize and eliminate tumors. Adjuvants are compounds that are necessary in cancer vaccines to mimic an infection process and amplify immune responses. The Toll-like receptor 2 and 6 (TLR2/TLR6) agonist dipalmitoyl-*S*-glyceryl cysteine (Pam_2_Cys) was demonstrated as an ideal candidate for synthetic vaccine adjuvants. However, the synthesis of Pam_2_Cys requires expensive *N*-protected cysteine as a key reactant, which greatly limits its application as a synthetic vaccine adjuvant in large-scaled studies. Here, we report the development of N-acetylated Pam_2_Cys analogs as TLR2/TLR6 agonists. Instead of *N*-protected cysteine, the synthesis utilizes *N*-acetylcysteine to bring down the synthetic costs. The *N*-acetylated Pam_2_Cys analogs were demonstrated to activate TLR2/TLR6 in vitro. Moreover, molecular docking studies were performed to provide insights into the molecular mechanism of how N-acetylated Pam_2_Cys analogs bind to TLR2/TLR6. Together, these results suggest *N*-acetylated Pam_2_Cys analogs as inexpensive and promising synthetic vaccine adjuvants to accelerate the development of cancer vaccines in the future.

## 1. Introduction

In the recent past, cellular immunotherapy has come up as one of the most suitable approaches for treating major diseases, such as infection and cancer, due to its high selectivity. The Chimeric Antigen Receptor T (CAR-T) therapy involves collecting T cells from patients and genetically modifying them to recognize specific antigens that are only expressed on tumor cells [1]. As immunotherapy approaches parallel to CAR-T therapy, cancer vaccines and synthetic vaccines utilize isolated or synthesized antigens to train the immune system to recognize tumors or pathogens [2]. The antigens used in cancer vaccines can be designed based on information collected from individual patients, thus opening up opportunities for personalized cancer treatment [3]. However, many isolated or synthetic antigens have poor immunogenicity on their own [4], and require adjuvants to help enhance the magnitude and quality of immune responses specific to various antigens [5]. 

Adjuvants include liposomes, lipopeptides, single stranded DNA etc., that mimic a natural infection to activate various immune components such as dendritic cells, macrophages and lymphocytes to produce desired immunological effects [6]. As a result, vaccine adjuvants can substantially reduce the number of requisite immunizations as well as the amount of antigen required [7]. The bacterial cell wall constituents Pam_2_Cys and Pam_3_Cys, together with the synthetic analogue Pam_2_CSK_4_ (Figure 1), have been shown as successful vaccine adjuvants due to their ability to activate Toll-like receptors (TLRs) [8,9,10]. Specifically, Pam_2_Cys was shown to be recognized by the toll-like receptor 2 (TLR2) and Toll-like receptor 6 (TLR6) heterodimers [11]. These adjuvants can induce TLR activation, which further activates NF-kB pathway on both innate and adaptive immune system to induce cytokine production and enhance immune responses against synthetic antigens [12]. 

Although the synthesis and many structural-activity relationship (SAR) studies of Pam_2_Cys and its analogues have been described [13,14,15,16,17], efficient large-scale synthesis of Pam_2_Cys analogues as synthetic vaccines adjuvants is still difficult due to its high cost. All previously reported synthetic methods involve costly synthetic strategies of orthogonal protection-deprotection techniques [13,14,15]. To ensure that the primary amine group on cysteine does not participate in any reactions during the synthesis, it has to be protected with either Boc [14,15] or Fmoc [13] protecting groups, which are cost inefficient for large scale synthesis. For the receptor-ligand interactions, an X-ray crystal structure of Pam_2_Cys bound to TLR2/TLR6 dimer suggests that the primary amine of the Cys residue does not have a major contribution to the ligand-receptor interactions through any hydrogen bonding interactions [11]. In addition, introducing N-acetylation to Pam_2_Cys analogs was shown to have minimal impact on the compound’s ability to induce immune responses [18]. 

In this research, novel synthetic pathways were developed using the inexpensive reactant *N*-acetylcysteine to synthesize *N*-acetyl Pam_2_Cys analogs, which can avoid orthogonal protection-deprotection steps and greatly reduce the costs for the synthesis of vaccine adjuvants. The ability of *N*-acetyl lipopeptides to activate TLR2/TLR6 signaling were examined in an NF-kB activation assay in comparison with the commercially available synthetic vaccine adjuvant candidate Pam_2_CysSK_4_ [8]. Molecular docking studies were performed to simulate the ligand-receptor interactions between *N*-acetyl Pam_2_Cys and TLR2/TLR6. Our results confirmed that N-acetyl Pam_2_Cys analogs can cause TLR2/TLR6 activation, and thus are promising candidates for cancer vaccine and synthetic vaccine adjuvants.

## 2. Results

### 2.1. Synthesis

Scheme 1 shows the novel synthetic design, which uses *N*-acetylcysteine instead of *N*-protected cysteine to avoid orthogonal protection-deprotection steps and lower synthetic costs. 1-hydroxyl and 2-hydroxyl groups on glycerol (**1**) were first protected by cyclohexanone (**2**) to form compound **3**. The hydroxyl group on compound **3** was charged by *p*-TsCl (**4**) to form compound **5**. The thiol group on N-acetylcysteine (**6**) reacts with compound **5**, yielding the thioether compound **7**. The two hydroxyl groups were first exposed by removing the protecting group with AcOH to yield compound **8**, and then coupled with fatty acids of different lengths to form the final products of **AHB 1–4** (Scheme 1, Table 1). The diastereomers were separated when they were first encountered in the synthesis i.e., from **6** to **7**. Product **7** used for further synthesis was diastereomerically pure. No further stereochemical complexity is observed thereafter. The AHB1-SK_4_ was further synthesized using standard Fmoc solid-phase peptide synthesis (Table 1).

### 2.2. In Vitro TLR2/6 Activation

It was previously reported that the acyl chain lengths of Pam_2_Cys analogs can affect the TLR activation [15]. To optimize the ability of N-acetyl lipopeptides to activate TLR2/TLR6, different analogs with varying diacyl chain lengths (12, 14, 16 or 18 carbons, Table 1) were synthesized. To test their abilities to activate TLR2/TLR6, TLR2 and TLR6 as well as an ELAM-SEAP reporter gene were transfected into HEK293 cells. Cells were treated with indicated compounds at 1 μM for 6 h, and the TLR2/TLR6 activation was tested by an NF-kB based activation assay. Specifically, TLR activation leads to the activation of the transcription factor NF-kB, which is recognized by the ELAM promoter to initiate the transcription/translation of the SEAP reporter. The SEAP protein is then quantified due to its ability to catalyze the hydrolysis of *p*-Nitrophenyl phosphate producing a yellow end product. *N*-acetyl lipopeptide with 18 carbons in the acyl chain (AHB-1) was shown to have the highest level of TLR2 and TLR6 activation (Figure 2A). To compare *N*-acetyl lipopeptides with the commercially available standard compound Pam_2_CSK_4_ [8] in the ability to activate the TLR2/TLR6, AHB-1 was conjugated with the short peptide SKKKK. The ability of AHB-1SK_4_ and Pam_2_CSK_4_ to activate TLR2/6 were tested in the NF-kB based activation assay. The results indicated that AHB-1SK_4_ was able to achieve 67% of the TLR2/TLR6 activation comparing to Pam_2_CysSK_4_ at 1 μM concentration (Figure 2B), suggesting that AHB-1 analogs can be effectively used as vaccine adjuvants through activating TLR2/6.

### 2.3. Docking Studies

To understand the effect of the *N*-acetyl group on lipopeptide binding to TLR2 and TLR6, in-silico molecular docking experiments were performed. The structure of Pam_2_CysSK_4_ bound to active TLR2 and TLR6 was previously determined [11] and was retrieved from Protein Data Bank (3A79). Since there are more than 50 rotatable bonds in Pam_2_CysSK_4_, which lead to inaccuracy in Glide flexible docking [19], the four lysine residues which do not interact with the receptor [20] were deleted and the diacyl chains were trimmed to six carbons to form the model structure of Cap_2_CysSer and N-acetyl Cap_2_CysSer molecules. The Cap_2_CysSer fits well into the binding pocket (Figure 3), with a Glide docking score of −8.149. With N-acetyl Cap_2_CysSer, a docking score of −10.025 was achieved, suggesting that the extra *N*-acetyl group does not abolish the interactions between lipopeptides and TLR2/6. What is more interesting, is that comparing to the original position of Cap2CysS, the N-acetyl Cap_2_CysS was found to be inserted deeper into the binding pocket (Figure 3). Our docking results suggest that the extra N-acetyl group can be tolerated during binding of N-acetyl Pam_2_Cys analogs to TLR2/6. 

## 3. Discussion

Though Pam_2_Cys analogs are widely considered as promising candidates for synthetic vaccine adjuvants [8], application in large scales can be limited by its relatively high synthetic costs. Here, we provide novel synthetic pathways to synthesize *N*-acetyl Pam_2_Cys analogs with inexpensive materials and avoiding extra orthogonal protection-deprotection steps. Our bioassay results confirmed that the N-acetyl Pam_2_Cys analogs can effectively activate TLR2/6, which is consistent with previous findings that an extra N-acetyl group did not produce any substantial difference in the ability of an Pam_2_CSK_4_ analog to induce CD80 expression [18]. Moreover, our docking results suggest that the extra N-acetyl group can be tolerated during ligand-receptor interactions. Given that the reported synthetic procedure of N-acetyl lipopeptides can effectively reduce the production cost, it is promising that N-acetyl lipopeptides will serve as adjuvants for synthetic vaccines in the future.

## 4. Materials and Methods 

### 4.1. Synthesis of (1,4-Dioxa-spiro[4,5]dec-2-yl)-methanol *(**3**)*



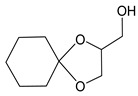



To a solution of glycerol 1 (5.0 g, 54.34 mmol) and cyclohexanone 2 (6.39 g, 65.16 mmol) in *n*-hexane (54.25 mL) was added conc. H_2_SO_4_ (3 mL) drop wise at 0 °C over a period of 15 min. The reaction mixture was slowly allowed to attain the room temperature and stirred for further 12 h. After completion of the reaction, as monitored by TLC, upper Hexane layer was separated. Then, powdered K_2_CO_3_ (1.41 g, 8.8 mmol) was charged for trapping the traces of acid that might be present in the organic layer. Finally, hexane layer was evaporated under vacuum to yield the crude product (10.0 g). The crude mixture was subjected to vacuum distillation to afford pure product 3 (8.41 g, 48.86 mmol, 89.9% yield) as a colorless syrupy liquid.

^1^H-NMR (CDCl_3_, 200 MHz): δ 1.41–1.64 (m,10 H), 3.55–3.61 (m, 1 H), 3.76–3.82 (m, 2 H), 3.99–4.07 (m, 1 H), 4.21–4.26 (m, 1 H). ESI-MS: 173 (M + H^+^). 

### 4.2. Synthesis of Toluene-4-sulfonic acid 1,4-dioxaspiro[4,5]dec-2-yl-methyl ester *(**5**)*



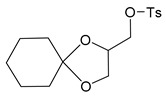



Compound **3** (8.41 g, 48.86 mmol) was dissolved in dry pyridine (35 mL) and immersed into an ice bath under nitrogen atmosphere. *p*-toluenesulfonyl chloride 4 (9.32 g, 48.0 mmol) was added slowly over a period of 20 min. to it and then the reaction mixture was slowly allowed to attain the room temperature and stirred for further 12 h. After completion of the reaction, as monitored by TLC, the reaction mixture was poured on to the crushed ice. The desired cyclohexanone protected glyceryl tosylate was obtained in its crude form after repeated extraction with water: EtOAc (4–25 mL). The crude product was purified using silica gel (100–200 mesh) chromatography to afford pure product 5as a colorless solid (10.95 g, 33.6 mmol, 70% yield).

^1^H-NMR (CDCl_3_, 200 MHz): δ 1.38–1.54 (m, 10 H), 2.46 (s, 3 H), 3.73–3.80 (m, 1 H), 3.92–4.12 (m, 3 H), 4.23–4.32 (m, 1 H), 7.37 (d, 2 H, *J* = 8.09 Hz), 7.81 (d, 2 H, *J* = 8.29 Hz). ESI-MS: 349 (M + Na^+^).

### 4.3. Synthesis of 2-Acetyl amino-3-(1,4-dioxa-spiro[4,5]dec-2-ylmethyl sulfanyl)-propionic acid *(**7**)*



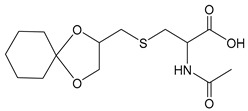



To the stirred suspension of N-acetyl cysteine 6 (5.47 g, 33.6 mmol) in methanolic KOH (2.81 g, 49.0 mmol in 40 mL methanol) solution, was added toluene-4-sulfonic acid 1,4-dioxa-spiro[4,5]dec-2-yl-methyl ester 5 (10.95 g, 33.6 mmol) under nitrogen at room temperature. Reaction mixture was heated at reflux for 8–10 h. After completion of the reaction, as monitored by TLC, reaction mixture was filtered off (potassium tosylate was precipitated during the course of reaction) and was acidified to 2 pH using 2 N HCl. Organic compound was extracted with DCM (3 × 100 mL) and Ethyl acetate (2 × 100 mL). The combined organic layers were dried over sodium sulphate and evaporated under vacuum to give the crude product which was purified on flash column chromatogram (silica gel as stationary phase, DCM: MeOH as mobile phase) to yield the product 7 in its pure form (8.2 g, 25.86 mmol, 77% yield).

^1^H-NMR (CDCl_3_, 200 MHz): δ 1.40–1.61 (m, 10 H), 2.05 (s, 3 H), 2.66–2.91 (m, 3 H), 3.29–3.31 (m, 1 H), 3.66–3.70 (m, 1 H), 4.03–4.11 (m, 1 H), 4.23–4.27 (m, 1 H), 4.59–4.63 (m, 1 H). ESI-MS: 340 (M + Na^+^).

### 4.4. Synthesis of 2-Acetyl amino-3-(2,3-dihydroxy-propyl sulfanyl)-propionic acid *(**8**)*



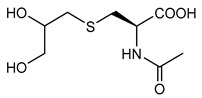



Compound 7 (8.2 g, 25.86 mmol) was dissolved in 75% aqueous acetic acid (20 mL) and heated to reflux for 3 h. After completion of the reaction, as monitored by TLC, the mixture was evaporated under vacuum to yield the crude product. The crude mass was purified through chromatography (Silica gel, 300–400 mesh) using chloroform: methanol gradient as the eluant to yield the pure compound 8 (5.80 g, 24.5 mmol, 94.6% yield).

^1^H-NMR (CDCl_3_, 200 MHz): δ 2.05 (s, 3 H), 2.68–2.93 (m, 3 H), 3.29–3.35 (m, 1 H), 3.68–3.76 (m, 1 H), 4.03–4.08 (m, 1 H), 4.27–4.30 (m, 1 H), 4.57–4.66 (m, 1 H). ESI-MS: 260 (M + Na^+^).

### 4.5. Synthesis of N-Acetyl Pam2Cys and Analogs *(**AHB-1–4**)*

To the stirred solution of compound 8 (1 g, 4.20 mmol, 1 equation), in TFA (Trifluoroacetic acid) (30 mL) was added palmitoyl chloride (2.45 g, 8.40 mmol, 2 eq.) slowly under nitrogen at room temperature. Reaction mixture was stirred for 30 min. After the completion of the reaction, as monitored by TLC, reaction mixture was dried under vacuum to yield the crude product (3.5 g), which was purified by flash chromatography (silica gel, 200–400 mesh) using chloroform and methanol gradient as eluant to afford *N*-Acetyl Pam2Cys (AHB2) in its pure form (2.69 g, 3.77mmol, 89.8% yield). The assignment of NMR peaks is given as under for AHB1 as the assignment will be same for all other compounds and only the number of methylene protons in the fatty chain will change. (**AHB2–4** do not show carbon numbers.) C1–17 in both the fatty chains are the same, thus we have given the same number to all such carbons in both the chains. Their protons and carbons have exactly the same chemical shift values.

Hexadecanoic acid 2-(2-acetylamino-2-carboxy-ethylsulfanyl)-1-octadecanoyl-oxy methyl–ethyl ester (*N*-Acetyl Str2Cys) (AHB1):



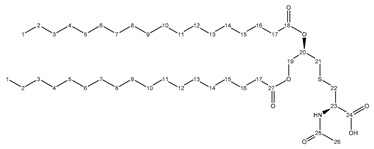



${\left[\alpha \right]}_{D}^{25}$ −15.2 (c 0.10, CHCl3), 1H-NMR (CDCl3, 400 MHz): 0.87–0.90 (t, 6H) [6Hs on two C1], 1.24–1.45 (m, 56H) [Protons on C2–15], 1.58–1.63 (m, 4H) [Protons on C16], 2.12 (s, 3H) [Protons on C27], 2.30–2.36 (m, 4H) [Protons on C17], 2.75 (m, 2H) [Protons on C21], 3.06–3.15 (m, 2H) [Protons on C22], 4.12–4.17 (m, 1H) [Proton on C19], 4.34–4.40 (m, 1H) [Next Proton on C19], 4.82 (m, 1H) [Protons on C23], 5.18 (m, 1H) [Protons on C20], 6.86–6.92 (m, 1H) [-NH proton] (Total assigned protons 82, Carboxyl proton out of range). 13C-NMR (CDCl3, 400 MHz): 14.12 [C1], 22.68 [C2], 24.88 [C26], 29.36, 29.48, 29.61[C3-C16 and C22], 31.90 [C17], 52.18 [C21], 52.51 [C23], 63.72 [C19], 70.25 [C20], 171.74 [C18], 172.58 [C27], 173.52 [C25], 173.66 [C24]. IR (KBr, cm^−1^): 3014, 2921, 2360, 1741, 1703, 1658, 1530, 1345, 1215, 1167, 964, 756. ESI-MS: 792 (M + Na+). Anal. Calcd. for C44H83NO7S, C, 68.61; H, 10.86; N, 1.82. Found C, 68.64; H, 10.83; N, 1.86.

Hexadecanoic acid 2-(2-acetylamino-2-carboxy-ethylsulfanyl)-1-hexadecanoyl-oxy methyl–ethyl ester (N-Acetyl Pam2Cys) (**AHB2**):



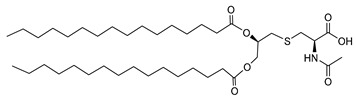



Specific rotation: ${\left[\alpha \right]}_{D}^{25}$ −21.2 (c 0.10, CHCl3), ^1^H-NMR (CDCl_3_, 400 MHz): δ 0.87–0.92 (m, 6H), 1.24–1.31 (m, 48H), 1.58–1.66 (m, 4H), 2.14 (s, 3H), 2.33–2.43 (m, 4H), 2.78 (d, 2H, *J* = 6.37 Hz), 3.10–3.17 (d, 2H, *J* = 54 Hz), 4.16–4.26 (m, 2H), 4.79 (m, 1H), 5.16 (m, 1H), 6.78–6.82 (m, 1H). ^13^C-NMR (CDCl_3_, 400 MHz): δ14.13, 22.70, 22.80, 24.71, 24.87, 29.12, 29.31, 29.38, 29.52, 29.71, 31.93, 32.93, 33.97, 34.11, 34.40, 52.04, 52.22, 63.72, 70.23, 70.27, 171.48, 172.99, 173.62, 173.69. IR (KBr, cm^−1^): 3013, 2921, 2851, 2360, 1741, 1703, 1658, 1530, 1464, 1452, 1374, 1345, 1215, 1167, 964, 755. ESI-MS: 713 (M + H^+^). Anal. Calcd. for C_40_H_75_NO_7_S, C, 62.28; H, 10.59; N, 1.96. Found C, 62.31; H, 10.55; N, 1.93.

Hexadecanoic acid 2-(2-acetylamino-2-carboxy-ethylsulfanyl)-1-tetradecanoyl-oxy methyl–ethyl ester (*N*-Acetyl Myr2Cys) (**AHB3**):



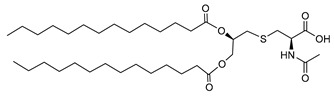



${\left[\alpha \right]}_{D}^{25}$ −27.7 (c 0.10, CHCl_3_), ^1^H-NMR (CDCl_3_, 400 MHz): δ 0.86–0.92 (t, 6H), 1.22–1.33 (m, 40H), 1.56–1.63 (m, 4H), 2.13 (s, 3H), 2.30–2.36 (m, 4H), 2.76 (m, 2H), 3.06–3.16 (m, 2H), 4.13–4.17 (m, 1H), 4.33–4.40 (m, 1H), 4.81 (m, 1H), 5.17 (m, 1H), 6.86–6.91 (m, 1H). ^13^C-NMR (CDCl_3_, 400 MHz): δ14.08, 22.66, 24.88, 29.36, 29.48, 29.61, 31.90, 52.18, 52.51, 63.72, 70.23, 70.27, 171.74, 172.52, 173.54, 173.67. IR (KBr, cm^−1^): 3012, 2921, 2851, 2360, 1741, 1701, 1658, 1530, 1464, 1452, 1376, 1345, 1215, 1167, 964, 754. ESI-MS: 680 (M + Na^+^), Anal. Calcd. for C_36_H_67_NO_7_S, C, 65.71; H, 10.26; N, 2.13. Found C, 65.82; H, 10.28; N, 2.10.

Hexadecanoic acid 2-(2-acetylamino-2-carboxy-ethylsulfanyl)-1-dodecanoyl-oxy methyl ethyl ester (*N*-Acetyl Lau2Cys) (**AHB4**):



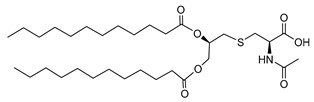



${\left[\alpha \right]}_{D}^{25}$ −31.3 (c 0.10, CHCl_3_), ^1^H-NMR (CDCl_3_, 400 MHz): δ 0.88–0.91 (t, 6H), 1.24–1.35 (m, 32H), 1.59–1.65 (m, 4H), 2.12 (s, 3H), 2.30–2.36 (m, 4H), 2.75 (m, 2H), 3.06–3.15 (m, 2H), 4.12–4.17 (m, 1H), 4.34–4.40 (m, 1H), 4.82 (m, 1H), 5.18 (m, 1H), 6.86–6.90 (m, 1H). ^13^C-NMR (CDCl_3_, 400 MHz): δ14.09, 22.68, 24.88, 29.36, 29.48, 29.61, 31.90, 52.18, 52.51, 63.72, 70.24, 70.27, 171.74, 172.58, 173.54, 173.68. IR (KBr, cm^−1^): 3014, 2921, 2851, 2360, 1741, 1703, 1658, 1530, 1464, 1452, 1376, 1345, 1215, 1167, 964, 756. ESI-MS: 601 (M + H^+^). Anal. Calcd. for C_32_H_59_NO_7_S, C, 63.86; H, 9.88; N, 2.33. Found C, 63.85; H, 9.83; N, 2.37.

### 4.6. Cell Culture

HEK293 cells were grown in the MEM medium (Minimum Essential Medium) (Gaithersburg, MD, USA) with 1sodium pyruvate (Sigma-Aldrich, St. Louis, MO, USA), 10% fetal bovine serum (Sigma-Aldrich, MO, USA) and 1% Pen-Strep (Sigma-Aldrich, MO, USA) at 37 °C in 5% CO_2_ incubator.

### 4.7. NF-kB Activation Assay

HEK293 cells (5 × 10^6^) were transiently transfected with 5.7 µg of ELAM-SEAP reporter gene (Invivogen, San Diego, CA, USA) and 0.3 µg of TLR2/TLR6 expression vector (Invivogen, CA) using the FuGENE 6 transfection kit (Roche Diagnostics, Mannheim, Germany). The cells were seeded into 96-well plates 6 h after transfection. After 24 h, the cells were treated with lipopeptides (1 µM) and the medium was collected 6 h after stimulation. SEAP activity was measure using SEAP reporter assay kit (Invivogen, CA) and µQuant Microplate Reader (BioTek, Winooski, VT, USA) at OD_412_.

### 4.8. Molecular Docking

The structure of the Pam_2_CSK_4_ bound to active TLR2 and TLR6 were retrieved from the 2.9Å crystal structure (PDB: 3A79) [20]. The TLR2/TLR6 structure was prepared by protein preparation wizards in Maestro with the force field of OPLS 2005. The ligand binding pocket was defined in the prepared receptor structure to generate a receptor grid by Glide receptor grid generation. The scaling factor was set to 1.0 and the partial charge cutoff was set to 0.25. The four lysine residues and 10 methylenes in each diacyl chain were deleted from the Pam_2_CSK_4_ structure to create the model of Cap_2_CS molecule. An acetyl group was added to the amide group of cysteine to create the model for N-acetyl Cap_2_CS. The structures of Cap_2_CS and *N*-acetyl Cap_2_CS were optimized using ligand preparation before docking. The ligand structures were docked into the receptor grid using Glide docking in flexible docking mode. The scaling factor was 0.8 and the partial charge cutoff was 0.15. The poses with the lowest docking score for each ligand were used for comparison. Poses were rejected if the Coulomb-vdW energy were greater than 0 kcal/mol. Extra precision (XP) was used in docking experiments.

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
