# Peer review of "Development of N-Acetylated Dipalmitoyl-S-Glyceryl Cysteine Analogs as Efficient TLR2/TLR6 Agonists"

_molecules, 2019, doi:10.3390/molecules24193512_

Round 1
Reviewer 1 Report
M. Cai et al. synthesized N-acetyl Pam2Cys derivatives and their O-stearyl as well as O-myristyl and -lauryl analogs as alternative adjuvants activating Toll-like receptors 2/6 (TLR2/6).
The biological evaluation was carried out through stimulation of HEK203 cells transfected with TLR2/6. The stearyl derivative AHB-1 exhibited about 2/3 of the activity of the natural Pam2Cys ligand. This is considered in interesting result because the synthesis of the N-acetyl cysteine derivatives is simpler than those of the components with unprotected amino groups.
In so far, the work is worth publishing.
However, the quality of the paper is not satisfying. The biological investigations (ELAM-SEAP?) are not explained (Figure 2). The chemical synthesis is also incompletely described. At what stage was the separation of diastereomers perfo4med on the way from 7 to the AHBs (Scheme 1)? The confirmation of the structure of the products is incomplete: Assignment of NMR signals is indispensable otherwise the data are useless.
In its present form the manuscript cannot be recommended for publication.
Author Response
Dear Reviewer,
Thanks for your comments and very important suggestions. Here we made corrections based on your comments:
We explained biological investigations (ELAM-SEAP?) (Figure 2).
Sorry for missing describing the separation of diastereomers (Scheme 1).
The diastereomers were separated when they were first encountered in the synthesis i,e from 6 to 7. Product 7 used for further synthesis was diastereomerically pure. No further stereochemical complexity is observed thereafter.
The products have been characterized through almost all the spectral techniques like 1H NMR, 13C NMR, Mass spectrometry, IR and elemental analysis. The structures are not that complex to go for sophisticated techniques like x-ray etc.
The assignment of NMR peaks is given as under for AHB-1 as the assignment will be same for all other compounds and only the number of methylene protons in the fatty chain will change:
We have assigned the 1H NMR and 13C peaks to all the protons and Carbons in the Spectrum of AHB-1. Most of the carbons in the 13C spectrum resonate at the same chemical shift and thus only few carbon signals are observed all of which have been assigned. C1-17 in both the fatty chains are the same, thus we have given the same number to all such carbons in both the chains. Their protons and carbons have exactly the same chemical shift values.
Reviewer 2 Report
The authors report herein the synthesis of a set of five Pam2Cys-related adjuvant molecules. The merit of the study resides on the use of a permanent N-acetylation of the cysteine during the curse of the synthesis. This simplifies the access tio this adjuvant family. Impact of this modification has been further checked on HEK293 expressing the TLR2/TLR6 heterodimer which is the receptor targeted by these adjuvants accompanied by molecular modeling simulations.
Lines 68-69 : the conclusion is rather abusive. AHB-1SK4 is only 67% as potent as Pam2CysSK4 (from Figure 2b). No comparison can be made for the other molecules since no positive control is reported in Figure 2a.
Regarding the docking studies, can the authors provide (with permission) the X-ray structure whereby both receptor and Pam2CysSK4 can be seen. It is not clear whether Cap2Cys adopts a position which can be superimposed with the natural adjuvant. Inother words is this position altered due to the absence of the large positively charged head and the truncation of the acyl chains ? That NAc Cap2Cys can be retrieved deeper within the binding pocket reflects the higher hydrophobicity of NAc compare to a free amine and suggests that the steric hindrance owing to the presence of the acetamido group does not impede with binding. We can hypothesize that if a simulation could be carried out with longer acyl chains, NAc Pam2Cys would be docked differently in the receptor. In conclusion, modeling does not preclude the binding of the NAc derivatives as suggested by their in vitro activity but does not bring many information regarding their actual binding and true comparison which, for example, might suggest further modification to synthesize more active compounds.
Minor comments
Could the uthor use react, couple rather than charge throughout the text?
Scheme 1 does not look very nice (e.g. TsCl drawing over the arrow) and must be improved; Figure 3 deserves to be enlarged to gain information.
Table 1: lipidated...activity
Line 98 [8] is underscripted,
Line 101 to achieve
One compound is sometimes referred to as Str2Cys. Does Str stands for Stearyl ? Is the abbreviation acknowledged by the community ?
No Capital letter for the naming of the compounds in the experimental section.
Line 157 p-toluenesulfonyl chloride
Line 195 chloroform
alpha D should be inserted as a formula.
Author Response
Dear Reviewer,
Thanks for your comments and very helpful suggestions! We made the following improvements based on your advice:
1. Line 68-69 was rewritten to avoid drawing conclusions that are abusive.
2. We will upload the docking results in mol2 format as supplementary material. Actually with the X-ray crystal structure of Pam2CSK4 bound to TLR2/6 (3A79), we tried to manually add an N-acetyl group to the Pam2CSK4 and it did not cause any steric clash, suggesting that the N-acetyl lipopeptides could possibly adopt the same conformation as Pam2CSK4 when bound to TLR2/6. We totally agree that our docking studies can't provide more valuable information to the ligand design due to having truncations, but the docking studies would fail to produce any results without truncations.
3. Scheme 1 and Figure 3 were revised to be more clear.
4. The arbitrary name of Str2Cys was removed.
5. Some grammar and formatting errors were fixed.
Round 2
Reviewer 1 Report
M. Cai et al. distinctly improved the manuscript. The biological evaluation is now sufficiently explained.
The stereochemistry of the synthesized products has been clarified, and the structure of the synthesized lipopeptides is now confirmed by assignment of the NMR signals.
The paper is now recommended for publication in Molecules.